# Mapping the Binding Sites of CA125-Specific Antibodies on a Revised Molecular Model of MUC16

**DOI:** 10.3390/cancers17091458

**Published:** 2025-04-26

**Authors:** Chien-Wei Wang, Anubhuti Srivastava, Eliza K. Hanson, Caitlin R. McEntee, Trisha Nair, Jane C. March, Rebecca J. Whelan

**Affiliations:** Department of Chemistry, University of Kansas, Lawrence, KS 66047, USA; chienwei@ku.edu (C.-W.W.); anubhuti@ku.edu (A.S.); hansone@rowan.edu (E.K.H.); caitlinmcentee@ku.edu (C.R.M.); tnair02@ku.edu (T.N.); janemarch@ku.edu (J.C.M.)

**Keywords:** ovarian cancer, CA125, MUC16, antibody, immunoassay, tandem repeat, ELISA, SPR

## Abstract

The serum CA125 test plays a crucial role in ovarian cancer care, with applications that include following the response to treatment and monitoring patients for recurrent disease. The CA125 test has limitations, however, displaying rates of false-positive and false-negative responses that preclude its use in population-wide screening. Importantly, the molecular-level mechanism underlying the test is poorly characterized, and the location and identity of the epitope(s) remain unknown. To identify the origin of false negative reporting in the CA125 test, we characterized the interactions between individual subdomains of CA125 and the antibodies used in the test. Building on a revised molecular-level description of MUC16 (the protein that carries the CA125 epitopes), we leverage mass spectrometry and multiple modes of binding characterization to demonstrate that the antibodies used in the CA125 test display discrepant, previously unknown, binding patterns. These insights reveal a mechanism underlying false negative results in serum CA125 testing and suggest new strategies for improving this important clinical assay.

## 1. Introduction

In 1981, Bast and co-workers reported the development of a monoclonal antibody, OC125, that could detect six ovarian carcinoma cell lines [1]. Two years later, they reported a radioimmunoassay to monitor ovarian cancer progression via testing serum levels of the antigen recognized by OC125. This antigen was named CA125 [2]. More than 40 years after these initial reports, CA125 is an FDA-approved biomarker to monitor the progression and recurrence of ovarian cancer and remains the gold standard biomarker for the clinical management of ovarian cancer [3]. Despite its crucial role in ovarian cancer care, many questions about the CA125 biomarker—and the assay that detects it—remain unanswered. CA125 is known to be a peptide epitope on the large mucin protein MUC16 [4,5]. MUC16 consists of a short membrane-spanning C-terminal region, a tandem repeat domain containing highly conserved but non-identical tandem repeats, and a large, heavily glycosylated N-terminus [6,7]. The tandem repeat region was originally proposed to contain 60 or more repeats [6], many of which were incompletely sequenced. Advances in long-read DNA sequencing technology recently enabled our group to completely sequence the tandem repeat region of MUC16 and propose a revised molecular model that contains 19 domains in the tandem repeat region [8]. With this advance, the composition of the tandem repeat region of MUC16 is now well characterized. However, the molecular-level identity of the CA125 biomarker remains an area of active investigation [9,10,11,12,13,14].

Serum CA125 levels are measured via the CA125 II test, a double-determinant immunoassay that uses OC125 and a second monoclonal antibody—M11—in a sandwich configuration [15,16]. While extensive evidence supports the hypothesis that OC125 and M11 bind within the tandem repeat region of MUC16, the CA125 epitopes have not been identified, and individually expressed tandem repeat proteins are not universally recognized by OC125 and M11 [10,11,13]. Determining the binding epitope(s) of OC125 and M11 may enable the development of alternative CA125 detection approaches, such as the use of a different combination of antibodies as capture and tracer, or the replacement of antibodies by alternative affinity agents such as aptamers or nanobodies [17]. Data-driven reinvention of the CA125 II test may improve its clinical utility by lowering the test’s limit of detection or enabling detection of MUC16 proteoforms that do not present the CA125 epitopes detected in the current assay [18].

We have previously published an initial survey characterizing the binding of four monoclonal CA125 antibodies (mAbs) to nine recombinant tandem repeat proteins individually expressed in *E. coli* [13]. That study used the numbering system of the MUC16 tandem repeat region derived from the 60+ tandem repeat model, which is now believed to be inaccurate. Here, we report an expanded analysis of mAb–tandem repeat binding based on our recently reported revised molecular model of MUC16 [8]. The present report studied the interaction of OC125, OC125-like, M11, and M11-like antibodies with 16 of 19 subdomains from the MUC16 tandem repeat region using two complementary affinity characterization methods: indirect enzyme-linked immunosorbent assay (ELISA) and localized surface plasmon resonance (SPR). Our previous work performed a semi-quantitative survey of binding affinity between mAbs and expressed tandem repeat proteins [13]. In the present work, we introduce a novel protocol using a digital microfluidics SPR platform to obtain binding kinetics as well as affinity [19,20]. Taken together, this study provides the most comprehensive characterization to date of the binding between expressed MUC16 tandem repeat domain proteins and the antibodies used in the CA125 II serum assay.

## 2. Materials and Methods

### 2.1. CA125 Recombinant Protein Expression and Purification

The sequence of the MUC16 tandem repeat region was obtained using long-read nanopore sequencing as previously reported [8]. The nucleotide sequences of 19 tandem repeats were synthesized and cloned into the pET-14b vector (GenScript, Piscataway, NJ, USA) using the XhoI and BamHI sites. Plasmid sequences were confirmed with Sanger sequencing by the vendor. Each plasmid was transformed into the SHuffle T7 Express *E. coli* strain (New England Biolabs, Beverly, MA, USA). The bacterial strains were grown in MagicMedia™ *E. coli* Expression Medium (Thermo, Waltham, MA, USA) with 100 μg/mL ampicillin at 30 °C to late stationary phase. Cells were harvested and lysed by freeze–thaw cycles and sonication in lysis buffer (20 mM sodium phosphate, 300 mM sodium chloride, 10 mM imidazole) with the cOmpletetTM protease inhibitor (Thermo, St. Louis, MO, USA). The 6-His-tagged recombinant protein was first purified using HisPur™ Ni-NTA Resin (Thermo, Waltham, MA, USA) according to the manufacturer’s protocol. The recombinant protein was further purified on an ÄKTA pure™ 25 chromatography system (Cytiva, Marlborough, MA, USA) containing a Superdex 75 10/300 GL column. The identity of each purified recombinant repeat protein was verified by western blotting with anti-His antibody and high-resolution mass spectrometry as described below. Protein purity was assessed by SDS-PAGE. Each recombinant repeat protein was independently expressed and purified at least three times.

### 2.2. Western Blot

The identity of each expressed tandem repeat protein was confirmed by western blotting. Purified recombinant repeat proteins were resolved by 16% sodium dodecyl sulfate-polyacrylamide gel electrophoresis at 100 V for 2 h and transferred to a polyvinylidene difluoride membrane. The membrane was blocked with 5% non-fat milk in TBST buffer (Tris-buffered saline with 0.05% Tween-20 from Fisher Scientific (Waltham, MA, USA)) at room temperature for 1 h and then incubated with anti-His primary antibody (Thermo, HIS.H8, 1:2000) at 4 °C overnight. The membrane was washed and incubated with anti-mouse HRP-conjugated secondary antibody (Thermo, 62-6520, 1:5000) at room temperature for 1 h. Pierce ECL Western Blotting Substrate (Thermo, Waltham, MA, USA) was used to visualize bands on an Azure 400 imager (Azure, Dublin, CA, USA).

### 2.3. Mass Spectrometry

#### 2.3.1. S-Trap Digestion 

The identity of each expressed tandem repeat protein was confirmed by mass spectrometry. Samples were digested using a protocol developed in our group for bottom-up proteomics analysis of MUC16 [21]. Sodium dodecyl sulfate (SDS), iodoacetamide (IAA), triethylammonium bicarbonate (TEAB), and Tris base were purchased from MilliporeSigma (St. Louis, MO, USA). Tris(2-carboxyethyl)phosphine (TCEP), deoxycholic acid (DCA), phosphoric acid, methanol (Burdick & Jackson), and sodium chloride (NaCl) were obtained from VWR (Radnor, PA, USA). Formic acid (99% purity; FA), acetonitrile (ACN), C_18_ ZipTips, and trypsin gold were purchased from Fisher Scientific (Hanover Park, IL, USA). S-Traps were purchased from Protifi (Huntington, NY, USA). A total of 30 µg of total protein was denatured and reduced by incubating in a solution of 100 mM TEAB containing 0.2% DCA, 7% SDS, and 10 mM TCEP for 10 min at 95 °C. Proteins were then alkylated by incubating with 10 mM IAA for 30 min in the dark at room temperature. Following alkylation, the reaction was quenched using 1.2% phosphoric acid. The protein solution was added to 100 mM TEAB in methanol and spun onto an S-Trap device. Samples were washed using 100 mM TEAB in methanol three times. Following washing, approximately 1.5 µg of trypsin dissolved in 100 mM TEAB in water was added to all devices. Samples were allowed to incubate for 6 h at 37 °C. The peptides were eluted in three steps using 100 mM TEAB in water, then 0.1% FA, and, finally, 50% ACN in 0.1% FA. Following elution, peptides were dried down to be desalted.

#### 2.3.2. In-Gel Digestion

The identity of bands resolved on SDS-PAGE gels of expressed tandem repeat proteins was investigated by mass spectrometry. Ten micrograms of total protein was loaded onto a 16% polyacrylamide gel and resolved at 150 V for 90 min. The gel was stained with Coomassie Brilliant Blue stain from Bio-Rad (Hercules, CA, USA). Bands located around the 20 kDa marker and below were separately excised from the gel and rinsed in 100 mM TEAB. A destaining cycle was performed until most of the stain was removed through dehydrating with a 2:1 ratio of ACN and 100 mM TEAB, followed by rehydration in 100 mM TEAB. To reduce the samples, the bands were dehydrated with 100% ACN, followed by rehydration in 100 mM TEAB and 10 mM TCEP for 20 min at 60 °C. The reduction solution was removed, replaced with 20 mM IAA, and left to incubate at RT for 20 min. The bands were rinsed with 100 mM TEAB, then put through the same destaining cycle as stated previously until all the stain was removed. A solution of 100% ACN was then added, and bands were dried in a Speedvac until all solution was removed. The bands were rehydrated in a solution that contained a 20:1 ratio of total protein to trypsin. The trypsin was immobilized in the gel by allowing the sample to incubate at 4 °C for 30 min. Digestion was allowed to occur at 37 °C overnight. Following digestion, the solution was removed and placed in a new tube. The bands were rinsed with 30% ACN and 0.1% FA for 5 min. This solution was combined with the solution in the new tube. The trypsin was then quenched using 10% FA. The peptide solution was then dried down to be desalted.

#### 2.3.3. Mass Spectrometry Data Acquisition

Peptides were resuspended in 0.1% FA and desalted using C18 ZipTips. The concentration of desalted peptides was determined through a fluorescent peptide BCA assay read in a SpectraMax M5 spectrophotometer (Molecular Devices, San Jose, CA, USA). Digested and desalted peptides were dried again and then resuspended in 2% ACN/0.1% FA. A total of 400 ng from each sample was analyzed using reversed-phase liquid chromatography on an UltiMate 3000 RSLCnano system (ThermoFisher Scientific, Waltham, MA, USA) using an in-house packed nano-LC column coupled to an Orbitrap Eclipse Mass Spectrometer (ThermoFisher Scientific, Waltham, MA, USA) operating in data-dependent mode.

#### 2.3.4. Mass Spectrometry Analysis

RAW data files were searched using Proteome Discoverer 3.1 (PD 3.1). The data files were searched against a *Homo sapiens* (Hs) database (downloaded from UniProt release 2024_01 on 9 April 2024) with the addition of individual tandem repeat sequences. Data were searched using oxidation of methionine residues (+15.995 Da), deamidation of asparagine, glutamine, and arginine (+0.984), N-terminal glutamate and glutamine conversion to pyroglutamate (−17.027 and −18.011 Da), N-terminal acetylation (+42.011 Da), and sodium adduction to aspartic and glutamic acid (+21.982 Da) as variable modifications and carboamidomethylation of cysteine residues (+57.021 Da) as a static modification. A false discovery rate of 1% was set for the peptide spectrum matches (PSMs). Data were analyzed and exported into Excel files.

### 2.4. ELISA

Pierce Nickel-Coated Plates (Thermo, Waltham, MA, USA) were used for indirect ELISA. An excess (885 ng) of His-tagged recombinant protein in 100 μL Phosphate Buffered Saline (PBS) buffer was incubated for 1 h at room temperature to saturate Ni-NTA binding sites. Saturation of the plates was confirmed in a separate experiment by varying the amount of His-tagged recombinant protein incubated over a range from 0 to 1600 ng (the theoretical capacity of the plates is 177 ng, according to the vendor). Anti-CA125 mAbs including anti-CA125 epitope group A1 (OC125, Sigma, diluted 1:200 and OC125-like clone M61704, Fitzgerald, 1:2000) and anti-CA125 epitope group B2/B1 (M11, Agilent, 1:100 and M11-like clone M61703, Fitzgerald, 1:2000) were used as primary antibodies. Anti-mouse HRP secondary antibody (Thermo, 62-6520, 1:20,000) was used to probe the immobilized primary antibodies. Both primary and secondary antibody incubations were at room temperature for 1 h. For the experiments varying anti-CA125 antibody incubation time, PBS was added to each well at the start of primary antibody incubation for 1 h at RT. During this incubation period, PBS was removed and anti-CA125 mAb (M11, Agilent, 1:500) was added to the wells at varied times to test the following incubation times: 0 min, 3 min, 10 min, 30 min, and 60 min. Chemiluminescence signals were developed using SuperSignal ELISA Femto Substrate (Thermo, Waltham, MA, USA) and detected on a SpectraMax M5 plate reader at 425 nm.

### 2.5. SPR Materials

All reagents loaded into SPR cartridges were supplemented with Tween 20 to a concentration of 0.1% (*v*/*v*). Glycine-hydrochloric acid (Glycine-HCl), sodium hydroxide (NaOH), sodium acetate, High Refractive Index Solution (32% glycerol), Low Refractive Index Solution (8% glycerol), Conditioning Solution (10 mM HCl), Quenching Solution (1 M ethanolamine, pH 8.5), anti-His immobilization kit, CBX carboxyl cartridges with cartridge fluid, 1-ethyl-3-(3-dimethylaminopropyl)carbodiimide hydrochloride (EDC), N-hydroxysuccinimide (NHS), and Alto Surface Plasmon Resonance Spectrometer were purchased from Nicoya Lifesciences (Kitchener, ON, USA). The Nanodrop 2000 was purchased from ThermoFisher (Waltham, MA, USA). PBS with Tween 20 (PBS-T) was made in-house with PBS from GrowCells (Irvine, CA, USA) supplemented with Tween 20. Water with Tween 20 (H_2_OT) was made in-house by adding 0.1% Tween 20 to 18.2 MΩ-cm purified water. Amicon Ultra-0.5 Centrifugal Filter Units were purchased from Fisher Scientific. Easy-Titer Mouse IgG Assay Kit was purchased from ThermoFisher. TraceDrawer Analysis 1.9.2 from Ridgeview Instruments (Uppsala, Sweden) was used to analyze SPR data.

### 2.6. M11 Stock Concentration Determination

As-purchased M11 was provided in liquid form in a buffer containing an unspecified amount of stabilizing protein and sodium azide. To determine the concentration of M11 in this reagent, the M11 solution was first buffer-exchanged using an Amicon Ultra 100 K MWCO filter (Millipore Sigma, Burlington, MA, USA) rinsed with MilliQ water. The mAb was buffer exchanged three times on the same column before recollecting in a fresh tube and resuspending in PBS-T at a volume approximately equivalent to the initial sample. M11 concentration was determined after buffer exchanges using the EasyTiter IgG assay (ThermoFisher Scientific, Waltham, MA, USA) with M11-like mAb prepared at known concentrations to generate a standard curve. The stock M11 concentration thus determined was used in preparing assay reagents.

### 2.7. SPR Binding Kinetics Assay Sample Preparation

A detailed description of the immobilization strategy and concentrations used in each test is reported in Appendix A. Direct immobilization of 6-His-tagged expressed tandem repeat proteins (hereafter, “repeats”) onto CBX cartridges was used for all repeats that immobilized above a threshold of 500 RU (measured after the quenching step). Repeats that did not successfully immobilize directly onto CBX cartridges were assayed via capture coupling on anti-His mAb cartridges. Repeat stock concentrations were measured on a NanoDrop using the Protein A_280_ mode with concentration estimated via the 1 Abs = 1 mg/mL setting. Proteins (repeats and antibodies) that were directly immobilized on CBX cartridges were prepared in sodium acetate at pH 5.5. Repeats that were assayed via capture coupling were diluted in PBS-T before loading into the cartridge, except in the case of low initial repeat concentration, when repeats were diluted in PBS and spiked with Tween 20 to ensure sufficient Tween concentration. Volumes recommended by Nicoya were used for loading the sample cartridges in the instrument, with 65 µL in row R, 4 µL in rows A and B, 2 µL in rows C-I, and 180 µL in row BF. For the anti-His cartridges prepared with the Nicoya immobilization kit, 3 µL was loaded in row C as per the kit recommendations.

### 2.8. SPR Instrumental Protocol

Experimental protocols were created using the Binding Kinetics application and the Direct Kinetics and Capture Kinetics assay templates in the Nicosystem. Titration was set to single-cycle. All experiments were performed on cartridges with CBX surface chemistry; repeats that required His immobilization bound anti-His mAb to the CBX cartridge surface via covalent coupling first and utilized the capture kinetics method. Sensors and reagents were held at 25 °C. All experiments used PBS-T as the running buffer. For the Activities settings, default times were used for startup, normalize, and clean. The build surface used either 300 or 900 s for the immobilization droplet time. The Direct Single Cycle Kinetics droplet times used default settings except for the Analyte Dissociation droplet, which was extended to 900 s, and the final baseline droplet, which was extended to 1000 s. For the anti-His capture cartridges, the pre-analyte baseline droplet, the dissociation droplet after analyte concentrations, and the final baseline droplet times were adjusted. For kinetics and affinity calculations, the molecular weights of repeats and antibodies were estimated to be 19 kDa and 150 kDa, respectively.

### 2.9. SPR Binding Kinetics Assessment

SPR data were automatically fit with a 1:1 binding model in the Nicosystem with calculations to report R_max_, *k*_a_, *k*_d_, and *K*_D_. A replicate was excluded as an outlier if it differed from the other values by two orders of magnitude or more. Data were downloaded, and calculated constants for each antibody were averaged across all non-excluded replicates. Interactions with R_max_ < 50 RU were considered non-binding when the repeats were directly immobilized, and interactions with R_max_ < 10 RU were considered non-binding when repeats were captured using an anti-His mAb. The difference in this cut-off can be attributed to higher repeat immobilization in direct coupling than in capture coupling. R_max_ is calculated by (MW analyte/MW of ligand) × RL, where R_max_ is the analyte binding signal when all ligand binding sites are occupied, and RL is the ligand immobilization level.

## 3. Results

### 3.1. Recombinant Tandem Repeat Protein Expression and Verification

The amino acid composition of the 19 MUC16 tandem repeats was obtained from our recently published report in which long-read sequencing was corroborated by mass spectrometry-based sequencing [8]. Each tandem repeat was recombinantly expressed in *E. coli* with an N-terminal 6-His tag and purified, first on Ni-NTA resin and then by high-performance gel filtration. Using this approach, 16 out of 19 repeats from the MUC16 tandem repeat region were obtained. Three repeats (R14, R18, and R19) consistently yielded insoluble protein despite extensive optimization attempts. These efforts included using different *E. coli* expression strains (SHuffle T7, NEB; ArcticExpress, Agilent, Santa Clara, CA, USA), evaluating various induction conditions (including Luria Broth with varying IPTG concentrations, MagicMedia Autoinduction Medium, and M9 minimal medium), and testing a range of expression temperatures (from 10 °C to 30 °C). Due to the inability to produce soluble protein, these three repeats were excluded from this study. Protein expression, purity, and molecular weight were verified with SDS-PAGE (image in Appendix A, densitometry data in Appendix A) and western blot with anti-His antibodies (image in Appendix A, densitometry data in Appendix A). All expressed tandem repeat proteins have a major band at the expected location (~19.7 kDa). The sequence of each recombinant repeat was verified with bottom-up proteomics on a high-resolution mass spectrometer, with sequence coverage ranging from 76% to 100% (Table 1 and Appendix A). The sequence alignment of the 19 tandem repeats is shown in Figure 1. Some blots show multiple bands; by gel excision and bottom-up proteomics analysis, these bands were confirmed to arise from proteoforms having the same amino acid composition within the center of each repeat, with identified peptides spanning positions 39–86. The bands that run lower on the gel may result from proteolytic clipping from the C-terminus during expression. The C-terminus lacks an attached tag (a 6-His tag is present on the N-terminus), and AlphaFold predicts that the C-terminal region of the tandem repeat proteins is relatively unstructured [8], making this part of the protein available for such proteolysis [22].

### 3.2. ELISA

The binding affinity between each tandem repeat protein and four anti-CA125 antibodies—M11, OC125, M11-like, and OC125-like—was measured via indirect ELISA. The repeat protein was immobilized on Ni-NTA ELISA plates and probed with anti-CA125 antibodies, followed by the HRP-conjugated anti-mouse secondary antibody. The capacity of the ELISA plate was determined by varying the concentration of R9, an exemplar repeat protein (Appendix A). The repeat protein concentration used in the ELISAs (885 ng) exceeded the theoretical capacity of the plate (177 ng) and is sufficient to saturate available binding sites on the plate. An unrelated protein (human epididymis protein 4, HE4) and no antigen served as negative controls. The binding pattern observed by ELISA is shown in Figure 2 (raw chemiluminescence data are reported in Appendix A). M11 was observed to bind all tandem repeats (panel A), while M11-like bound to all repeats except R1 (panel C). M11 exhibited negligible binding to both negative controls (no antigen and an unrelated protein, HE4), whereas M11-like exhibited high nonspecific signal in the no-antigen control, likely owing to nonspecific binding to the ELISA plate, which did not occur in the HE4 control or in R1, where the ELISA plate was saturated with His-tagged protein. OC125 bound 11 of the 16 repeats tested (R1, R4, R5, R7, R8, R9, R11, R13, R15, R16, and R17; panel B), while the OC125-like antibody bound to all repeats except R11 and R17 (panel D). Negligible binding to the negative controls was observed for both OC125 and OC125-like mAbs. Differences between antibody–tandem repeat combinations categorized as “binding” and those categorized as “non-binding” were compared using Welch’s *t*-test assuming unequal variances. For OC125, M11-like, and OC125-like, the differences between binding and non-binding were highly significant, with *p* < 0.0001 at 95% confidence.

### 3.3. SPR

The binding affinity and kinetics of the interaction between each tandem repeat protein and four anti-CA125 antibodies—M11, OC125, M11-like, and OC125-like—were measured via SPR. Repeat proteins were immobilized onto the sensing surface either by direct coupling onto a carboxyl surface using an EDC-NHS reaction with primary amines on the repeats or by capture coupling on an anti-His mAb-functionalized surface. For each antibody–tandem repeat combination, the SPR assay returned binding kinetics data (*k*_a_ and *k*_d_) and binding affinity data (*K*_D_, calculated from *k*_d_/*k*_a_). The results of SPR binding kinetics experiments are shown in Figure 3, and individual SPR binding parameters are presented in Appendix A (*k*_a_), Appendix A (*k*_d_), and Appendix A (*K*_D_). For some antibody–repeat combinations—those with slow dissociation kinetics—the fit to the experimental data yielded *k*_d_ values too small to be confidently calculated. In these cases, a default value of *k*_d_ = 1.00 × 10^−6^ M^−1^s^−1^ is given. The corresponding *K*_D_ values should be understood to be estimates. In SPR assays, M11 was observed to bind to 12 of the 16 repeats tested (R1, R2, R3, R5, R7, R8, R10, R11, R13, R15, R16, and R17; panel A). OC125 displayed binding to 9 of the 16 repeats tested (R1, R4, R5, R7, R8, R11, R15, R16, R17; panel B). Reliable binding data could not be collected for R9 and R13 with OC125 due to high non-specific binding observed in these cases, which are labeled “NA.” M11-like was observed to bind to all repeats except R1 (panel C). OC125-like was observed to bind all repeats except R11 and R17 (panel D).

### 3.4. Comparison of Analytical Methods (ELISA and SPR)

Figure 4 summarizes the antibody–tandem repeat binding pattern determined by SPR and ELISA. Combinations for which binding was observed are labeled with a check; non-binding combinations are represented by “X”. For SPR, depending on the repeat immobilization strategy, interactions with R_max_ < 50 RU (direct coupling) and R_max_ < 10 RU (capture coupling) were considered non-binding. For ELISA, interactions that resulted in chemiluminescent signals less than 20% of the highest signal observed for antibody–tandem repeat binding within the dataset were classified as non-binding. On balance, the two methods of affinity characterization agree. The pattern of binding for OC125, M11-like, and OC125-like is in complete agreement between the two methods. For M11, the ELISA data reveal binding to all 16 of the tandem repeats studied, whereas the SPR data do not show M11 binding to R4, R6, R9, and R12. Differences in response between SPR and ELISA have been previously reported. A study by Heinrich et al. exploring the differences observed between ELISA and SPR measurements of kinetic parameters for an antibody/antigen complex attributed the differences to two different conformations of the complex with different dissociation constants that arises from binding occurring in two steps (an initial low-affinity binding of proteins followed by a conformational change to a more stable, higher affinity complex) [23]. In addition to the contact time being potentially insufficient to capture enough analyte, it is unknown whether the two methods have different ligand densities immobilized and whether the conditions at the surface cause steric hindrance. Serrano and co-authors compared ELISA data to SPR analysis on the extent of protein holotoxin disassembly, observing the same overall trend between the techniques but differences in the completeness of disassembly [24]. They argue that the flow of running buffer in SPR introduces shear forces that are not present in an ELISA plate. The difference in physical conditions during incubation could also affect the speed or extent to which the analyte measurably binds to the ligand of interest. We hypothesized that this difference—M11 is observed to bind all repeats when tested via ELISA, while SPR shows that M11 recognizes fewer repeats—was attributable to the relatively short contact time in SPR experiments being insufficient to capture enough M11 onto the SPR surface to reach the threshold. To test this hypothesis, we immobilized R4, R6, R9, and R12 on ELISA plates for 1 hr, then incubated M11 for varying amounts of time ranging from 0 min to 60 min. The normalized ELISA data in Figure 5 show that, at 3 min (the amount of time during which M11 would be in contact with the sensing surface in an SPR experiment), the signal is significantly less than at later time points, supporting the hypothesis. In light of our data demonstrating the significant effect of contact time on ELISA measurements, we conclude that the two methods of analysis are in complete agreement with each other and that M11 binds to all repeats tested.

## 4. Discussion

In the four decades since the initial report of CA125 as an ovarian cancer biomarker, the binding between CA125-specific antibodies and the CA125 epitope(s) has remained incompletely characterized. In prior work [13], we expressed nine proteins from the tandem repeat domain of MUC16 (the proposed location of the CA125 epitopes) and characterized the binding of OC125, M11, OC125-like (clone M61704), and M11-like (clone M61703) to the expressed repeats using three methods of affinity characterization: western blotting, ELISA, and SPR. That study used the numbering system based on the foundational publication by O’Brien and co-workers [6] and predates our 2024 reporting of a revised molecular model of the MUC16 tandem repeat region containing 19 subdomains rather than 63 [8]. Because the current study uses the revised molecular model we reported in 2024, some of the repeats characterized in our 2023 report—specifically, what we now refer to as “old R6”, “old R25”, and “old R34”—are not part of the current study. It is notable that “old R25” has 98% sequence identity to R5 in the 153 residues where they overlap (out of 156 amino acids total). Studies (ours and others) that reported on “R25” using the old numbering system can be directly compared to R5 in the revised MUC16 model. Sequence similarities between the old and new numbering systems can be found in our prior work [8].

The patterns of binding reported here align with the results obtained in our previous study. In this study, M11 was observed to bind to all tandem repeats tested. In our 2023 report, M11 was observed by ELISA and SPR to bind to all repeats tested except old R6, which is not included in the revised molecular model and was therefore not studied. The binding of OC125 is more complicated, but all the repeats that were interrogated both in this study and our earlier work display the same pattern, namely, that R5 (old R5, same as old R25), R8 (old R9), and R9 (old 11) bind OC125, whereas R2 (old R2) and R6 (old R7) do not. For the M11-like clone, the only repeat found in this study to not bind is R1, which was not included in our earlier report, while all the other repeats that are in common—R2 (old R2), R5 (old R5), R6 (old R7), R8 (old R9), and R9 (old R11)—display binding. For the OC125-like clone, we found, in this study, that all repeats except R11 and R17 bind. Neither of these was studied in our 2023 report.

The most comprehensive study prior to 2023 investigating the location of the CA125 epitopes was conducted by Bressan and co-workers [10], who used western blot analysis of six recombinant CA125 repeats, which they referred to as R2, R7, R9, R11, R25, and R51, using the O’Brien numbering system. Using the revised molecular model, we renumber the repeats interrogated in the Bressan study as R2, R6, R8, and R9. As mentioned earlier, old R25 shares nearly 100% sequence overlap with R5, and R51 is not included in the revised molecular model. Bressan and co-workers observed that all repeats they expressed bind M11, consistent with the pattern reported here. They observed strong binding of old R25 and old R11 (R5 and R9 in our numbering system) to OC125, while old R2 and old R7 (R2 and R6 in our numbering system) display negligible binding to OC125: Bressan’s results are in congruence with what we report here. An elegant study by Clausen and co-workers [11] studied CA125 mAb binding to various truncation constructs centered on tandem repeat 5, with the largest construct comprising the SEA5 domain and linker regions, the repeats on either side (R4 and R6). Although the expressed proteins studied here do not directly correlate with the set studied by Clausen, it is notable that they observed OC125 and M11 reacted with R5-containing constructs in SDS-PAGE western blot analysis, consistent with our ELISA and SPR data.

The possible connections between the differential binding reported here and clinical impacts are multifold. The first connection relates to splice variants and their role in false negative reporting of the CA125 assay. In our 2024 publication presenting the revised molecular model of MUC16, we reported the observation of possible splice variants in MUC16 mRNA from a high-grade serous ovarian tumor. The variants lacked entire subunits within the tandem repeat domain. Depending on the MUC16 proteoforms expressed by an individual, the differential binding documented here could lead to false negatives in clinical testing, because both OC125 and M11 must successfully and simultaneously bind to CA125 for the assay to generate a signal. The presence of MUC16 splice variants in ovarian cancer and other cancers is still an area of active investigation, but some intriguing evidence suggests that mucin splice variants are a hallmark of cancer. Thompson et al. used publicly available RNA sequencing datasets to assess the expression of mucin family members and their splice variants in pancreatic ductal adenocarcinoma and found that patients expressing MUC16 splice variants had shorter survival [25]. Rao et al. found that fibroblast cells engineered with truncations of MUC16 remained able to form tumors and retained metastatic properties [26]. Intriguingly, He and co-workers showed that epithelial splicing regulatory protein (ESRP2), which is involved in the generation of splice variants, is highly expressed in breast cancer and correlated with poor prognosis in these patients. ESRP2 is also upregulated in ovarian cancer [27].

The second connection to clinical impacts lies in the design of the CA125 test itself. The original version of the test used OC125 as both capture and tracer. First-generation CA125 kits were found to give discrepant or discordant results [28,29], which motivated the development of an improved assay. The current version of the test is the second generation, or “CA125 II”, with M11 and OC125 serving roles of capture and tracer, respectively (the roles are reversed in some versions of the test) [30]. The second-generation test was found to have superior analytical performance to the first-generation test [30]. Given the differential binding pattern reported here, with M11 binding to tandem domain repeats more consistently than OC125, the exciting possibility presents itself of using M11 as both capture and tracer; work towards this “CA125 III” test is actively ongoing in our laboratory.

We note that a few aspects of the experimental design may limit the generality of the findings reported here. First, our approach could be characterized as reductionistic, expressing individual domains from the tandem repeat region as recombinant proteins and characterizing their binding to anti-CA125 mAbs. This approach is, however, consistent with the precedent set by other researchers who have endeavored to locate the CA125 epitopes [10,11] and is suitable given the extremely large size (3–5 MDa) and complexity of native MUC16. Second, we use an *E. coli* expression system, which does not recapitulate mammalian post-translational modification such as glycosylation. We argue, however, that characterizing an expressed tandem repeat protein without native glycosylation still offers valuable insight on mAb binding for two main reasons: in foundational work by Davis et al. characterizing the CA125 antigen, binding of OC125 was found to be independent of glycosylation [31], and, in more recent work by Marcos-Silva et al., the pattern of mAb binding to expressed subdomains of MUC16 did not differ between proteins expressed in *E. coli* and proteins expressed in Chinese Hamster Ovary cells [11]. This prior finding supports the validity of the expression strategy used here.

## 5. Conclusions

The recent use of long-read sequencing to obtain a revised molecular model of the tandem repeat domain of MUC16 [8] has enabled the binding pattern of CA125-specific antibodies to be characterized with previously unattainable coverage and accuracy. The observations of greatest potential clinical significance relate to M11 and OC125, the antibodies that comprise the sandwich immunoassay used for clinical management of ovarian cancer. From the studies reported here, we conclude that M11 binds uniformly across the tandem repeat domain of MUC16 (this conclusion is tempered by the caveat that only 16 of 19 tandem repeat domains were included in the present work). In contrast, OC125 binds only 11 of the repeats studied, and does not display binding to the other five (again, conclusions cannot be drawn about the three repeats that were not studied). Considering that binding of both OC125 and M11 is required for the generation of a signal in the clinical immunoassay, this differential binding pattern may present a mechanism of undercounting MUC16 present in the serum of ovarian cancer patients. The generation of alternative affinity agents and alternative assays is an active area of research in our laboratory. Finally, we note that this nearly complete binding pattern contributes to ongoing efforts to identify the CA125 epitopes with amino-acid level precision.

## Figures and Tables

**Figure 1 cancers-17-01458-f001:**
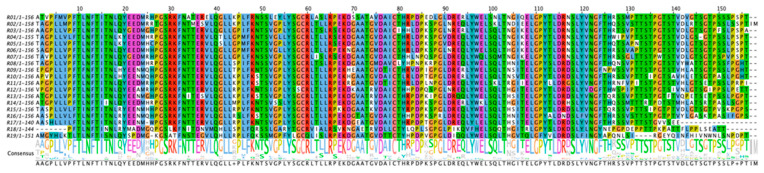
Clustal Omega (version 1.2.2) alignment of the 19 tandem repeats of MUC16. The tandem repeats were aligned using Clustal Omega and visualized using Jalview (version 2.11.4.0) with the Clustal Omega standard color scheme. The consensus sequence shows amino acid abundance at each position.

**Figure 2 cancers-17-01458-f002:**
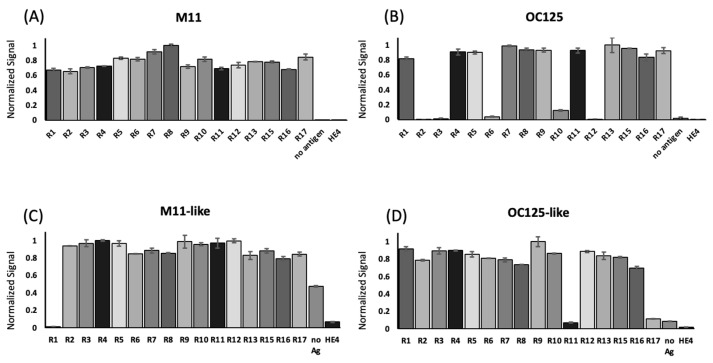
Results of indirect ELISA. Individually expressed recombinant repeat proteins from the tandem repeat region of MUC16 were probed with (**A**) M11 (at a dilution of 1:100), (**B**) OC125 (1:200), (**C**) M11-like (M61703, 1:2000), and (**D**) OC125-like (M61704, 1:2000). Chemiluminescent signals were normalized (to max = 1.0) within each data set. No antigen and HE4 were used as negative controls. Error bars are the standard error of the mean (*n* = 3). Welch’s t-test for unequal variances was applied to compare binding and non-binding repeat–antibody combinations for OC125, M11-like, and OC125-like antibodies. At 95% confidence, the differences between binding and non-binding were highly significant (*p* < 0.0001).

**Figure 3 cancers-17-01458-f003:**
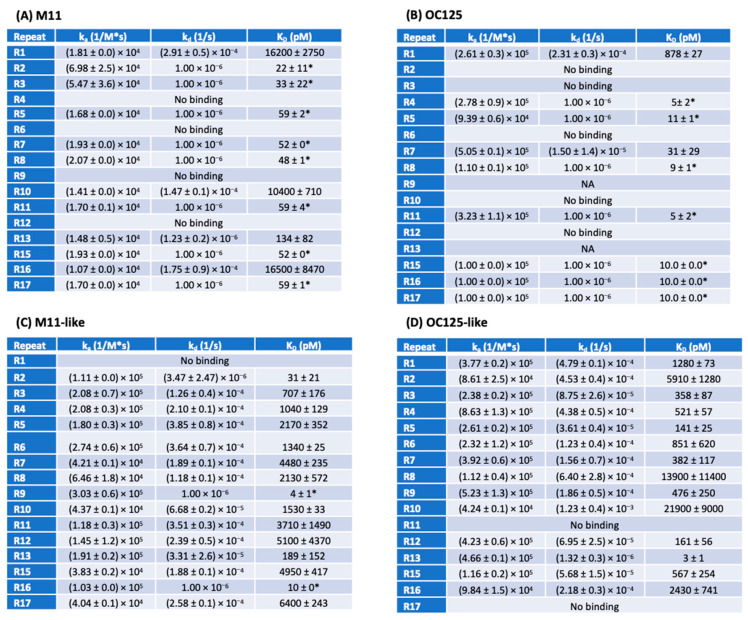
Results of SPR binding kinetics experiments between expressed tandem repeat proteins and the CA125-specific antibodies (**A**) M11, (**B**) OC125, (**C**) M11-like, and (**D**) OC125-like. Depending on the tandem repeat immobilization strategy, interactions with R_max_ < 50 RU (direct coupling) and R_max_ < 10 RU (capture coupling) are classified as “No binding”. When the dissociation rate constant (*k*_d_) was too slow to be determined experimentally, the default value of *k*_d_ = 1.00 × 10^−6^ was used, and the *K*_D_ (represented with an *) is an estimated value that represents an upper limit. All *K*_D_ values are reported as standard error of the mean (SEM) with *n* = 3 for most experiments. Replicate numbers for each tandem repeat protein–antibody experiment are reported in Appendix A.

**Figure 4 cancers-17-01458-f004:**
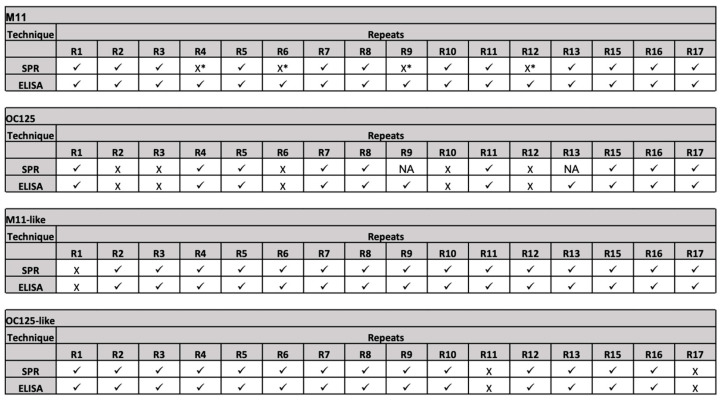
Antibody–tandem repeat protein binding pattern observed in SPR and ELISA. Antibody–repeat combinations for which interaction was observed are labeled with a check. For SPR, depending on the repeat immobilization strategy, antibody–repeat interactions with R_max_ < 50 RU (direct coupling) and R_max_ < 10 RU (capture coupling) were manually categorized as non-binding (represented by “X”). For ELISA, interactions with chemiluminescent signal less than 20% of the highest signal observed for the antibody–repeat binding within a dataset were classified as non-binding (represented by “X”). Non-binding interactions of M11 with R4, R6, R9, and R12 are labeled with “X*” to reflect the observation that, for these combinations, the antibody contact time was likely insufficient, and that the interaction observed in ELISA experiments is real. The interaction of R9 and R13 with OC125 could not be determined because of high non-specific binding in these cases and is labeled “NA”.

**Figure 5 cancers-17-01458-f005:**
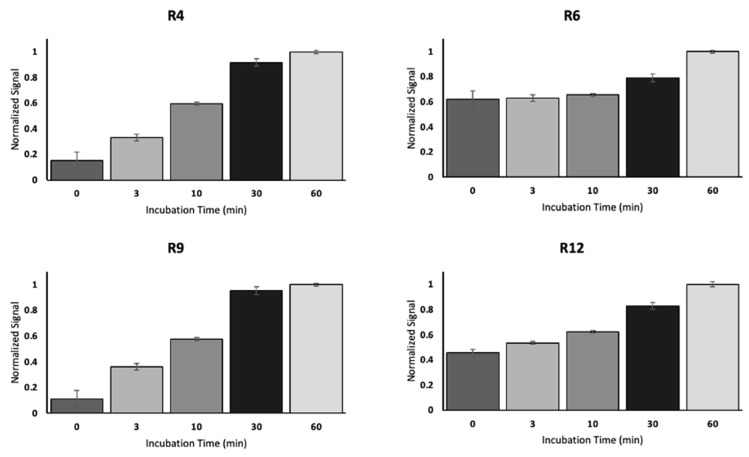
Results of ELISA experiments in which expressed MUC16 tandem repeat proteins (R4, **top left**; R6, **top right**; R9, **bottom left**, and R12, **bottom right**) were immobilized via 6-His tags for 1 h followed by incubation with anti-CA125 antibody M11 for varying times ranging from 0 min to 60 min. The pattern of signal growth reveals that 3 min of incubation with M11 (the contact time used in SPR experiments) is not sufficient to achieve saturation of the binding surface. Error bars are standard deviation (*n* = 3). Differences between 3 min and 60 min were found to be significant by a one-tailed t-test at 95% confidence, with *p* = 0.0001 for R4, *p* = 0.0006 for R6, *p* = 0.00005 for R9, and *p* = 0.0002 for R12.

**Table 1 cancers-17-01458-t001:** Amino acid sequences of the 19 domains in the MUC16 tandem repeat region. The number of each tandem repeat is in the left column, the amino acid sequence is in the center, and the percent coverage of each expressed repeat obtained from bottom-up proteomics analysis is in the right column. Three of the repeats (R14, R18, and R19) were not obtained in soluble form and are not included in this study.

Repeat	Sequence	Coverage
R1	ATVPFMVPFTLNFTITNLQYEEDMRHPGSRKFNATERELQGLLKPLFRNSSLEYLYSGCRLAS LRPEKDSSATAVDAICTHRPDPEDLGLDRERLYWELSNLTNGIQELGPYTLDRNSLYVNGFT HRSSMPTTSTPGTSTVDVGTSGTPSSSPSPT	100%
R2	TAGPLLMPFTLNFTITNLQYEEDMRRTGSRKFNTMESVLQGLLKPLFKNTSVGPLYSGCRLT LLRPEKDGAATGVDAICTHRLDPKSPGLNREQLYWELSKLTNDIEELGPYTLDRNSLYVNGF THQSSVSTTSTPGTSTVDLRTSGTPSSLSSPTIM	100%
R3	AAGPLLVPFTLNFTITNLQYGEDMGHPGSRKFNTTERVLQGLLGPIFKNTSVGPLYSGCRLTS LRSEKDGAATGVDAICIHHLDPKSPGLNRERLYWELSQLTNGIKELGPYTLDRNSLYVNGFT HRTSVPTTSTPGTSTVDLGTSGTPFSLPSPA	97%
R4	TAGPLLVLFTLNFTITNLKYEEDMHRPGSRKFNTTERVLQTLLGPMFKNTSVGLLYSGCRLT LLRSEKDGAATGVDAICTHRLDPKSPGVDREQLYWELSQLTNGIKELGPYTLDRNSLYVNGFT HWIPVPTSSTPGTSTVDLGSGTPSSLPSPT	99%
R5	TAGPLLVPFTLNFTITNLKYEEDMHCPGSRKFNTTERVLQSLLGPMFKNTSVGPLYSGCRLT LLRSEKDGAATGVDAICTHRLDPKSPGVDREQLYWELSQLTNGIKELGPYTLDRNSLYVNG FTHQTSAPNTSTPGTSTVDLGTSGTPSSLPSPT	100%
R6	SAGPLLVPFTLNFTITNLQYEEDMHHPGSRKFNTTERVLQGLLGPMFKNTSVGLLYSGCRLT LLRPEKNGAATGMDAICSHRLDPKSPGLNREQLYWELSQLTHGIKELGPYTLDRNSLYVNG FTHRSSVAPTSTPGTSTVDLGTSGTPSSLPSPT	95%
R7	TAVPLLVPFTLNFTITNLQYGEDMRHPGSRKFNTTERVLQGLLGPLFKNSSVGPLYSGCRLIS LRSEKDGAATGVDAICTHHLNPQSPGLDREQLYWQLSQMTNGIKELGPYTLDRNSLYVNG FTHRSSGLTTSTPWTSTVDLGTSGTPSPVPSPT	100%
R8	TAGPLLVPFTLNFTITNLQYEEDMHRPGSRKFNTTERVLQGLLSPIFKNSSVGPLYSGCRLTSL RPEKDGAATGMDAVCLYHPNPKRPGLDREQLYWELSQLTHNITELGPYSLDRDSLYVNGF THQNSVPTTSTPGTSTVYWATTGTPSSFPGHT	100%
R9	EPGPLLIPFTFNFTITNLHYEENMQHPGSRKFNTTERVLQGLLTPLFKNTSVGPLYSGCRLTLL RPEKHEAATGVDTICTHRVDPIGPGLDRERLYWELSQLTNSITELGPYTLDRDSLYVNGFNP WSSVPTTSTPGTSTVHLATSGTPSSLPGHT	100%
R10	APVPLLIPFTLNFTITNLHYEENMQHPGSRKFNTTERVLQGLLKPLFKSTSVGPLYSGCRLTLL RPEKHGAATGVDAICTLRLDPTGPGLDRERLYWELSQLTNSVTELGPYTLDRDSLYVNGFT HRSSVPTTSIPGTSAVHLETSGTPASLPGHT	100%
R11	APGPLLVPFTLNFTITNLQYEEDMRHPGSRKFNTTERVLQGLLKPLFKSTSVGPLYSGCRLTL LRPEKRGAATGVDTICTHRLDPLNPGLDREQLYWELSKLTRGIIELGPYLLDRGSLYVNGFT HRNFVPITSTPGTSTVHLGTSETPSSLPRPI	100%
R12	VPGPLLVPFTLNFTITNLQYEEAMRHPGSRKFNTTERVLQGLLRPLFKNTSIGPLYSSCRLTLL RPEKDKAATRVDAICTHHPDPQSPGLNREQLYWELSQLTHGITELGPYTLDRDSLYVDGFT HWSPIPTTSTPGTSIVNLGTSGIPPSLPETT	100%
R13	ATGPLLVPFTLNFTITNLQYEENMGHPGSRKFNITESVLQGLLKPLFKSTSVGPLYSGCRLTLL RPEKDGVATRVDAICTHRPDPKIPGLDRQQLYWELSQLTHSITELGPYTLDRDSLYVNGFTQ RSSVPTTSTPGTFTVQPETSETPSSLPGPT	88%
R14	ATGPVLLPFTLNFTIINLQYEEDMHRPGSRKFNTTERVLQGLLMPLFKNTSVSSLYSGCRLTL LRPEKDGAATRVDAVCTHRPDPKSPGLDRERLYWKLSQLTHGITELGPYTLDRHSLYVNGF THQSSMTTTRTPDTSTMHLATSRTPASLSGPT	N/A
R15	TASPLLVLFTINFTITNLRYEENMHHPGSRKFNTTERVLQGLLRPVFKNTSVGPLYSGCRLTL LRPKKDGAATKVDAICTYRPDPKSPGLDREQLYWELSQLTHSITELGPYTLDRDSLYVNGF TQRSSVPTTSIPGTPTVDLGTSGTPVSKPGPS	99%
R16	AASPLLVLFTLNFTITNLRYEENMQHPGSRKFNTTERVLQGLLRSLFKSTSVGPLYSGCRLTL LRPEKDGTATGVDAICTHHPDPKSPRLDREQLYWELSQLTHNITELGPYALDNDSLFVNGF THRSSVSTTSTPGTPTVYLGASKTPASIFGPS	76%
R17	AASHLLILFTLNFTITNLRYEENMWPGSRKFNTTERVLQGLLRPLFKNTSVGPLYSGCRLTLL RPEKDGEATGVDAICTHRPDPTGPGLDREQLYLELSQLTHSITELGPYTLDRDSLYVNGF THRSSVPTTSTGVVSEE	100%
R18	PFTLNFTINNLRYMADMGQPGSLKFNITDNVMQHLLSPLFQRSSLGARYTGCRVIALRSVK NGAETRVDLLCTYLQPLSGPGLPIKQVFHELSQQTHGITRLGPYSLDKDSLYLNGYNEPGPD EPPTTPKPATTFLPPLSEATT	N/A
R19	AMGYHLKTLTLNFTISNLQYSPDMGKGSATFNSTEGVLQHLLRPLFQKSSMGPFYLGCQLIS LRPEKDGAATGVDTTCTYHPDPVGPGLDIQQLYWELSQLTHGVTQLGFYVLDRDSLFINGY APQNLSIRGEYQINFHIVNWNLSNPDPT	N/A

## Data Availability

Data are available upon request to interested researchers.

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
