# Peer review of "Mapping the Binding Sites of CA125-Specific Antibodies on a Revised Molecular Model of MUC16"

_cancers, 2025, doi:10.3390/cancers17091458_

Round 1
Reviewer 1 Report
Comments and Suggestions for Authors
The manuscript presents a thorough study on the binding interactions of CA125-specific antibodies with MUC16 tandem repeat proteins. The work is well-structured, methodologically sound, and provides significant insights into the molecular-level interactions that underlie the CA125 immunoassay. The experimental design, using a combination of ELISA and SPR, is appropriate for characterizing antibody-epitope binding. The study's conclusions have potential implications for improving ovarian cancer diagnostics.
- The manuscript includes appropriate controls and replicates, but additional statistical analyses (e.g., confidence intervals for KD values in SPR) would provide more robust conclusions.
- Inconsistencies between ELISA and SPR for OC125 and M11-like antibodies should be discussed more explicitly beyond differences in contact time
- The findings indicate that OC125 may not bind to all MUC16 repeats, suggesting the current CA125 assay might under-detect certain MUC16 proteoforms.
- It would be beneficial to discuss whether specific tandem repeat domains correlate with ovarian cancer progression or prognosis, linking molecular findings to clinical implications.
- Figures showing ELISA and SPR results are useful, but adding error bars and statistical significance indicators would improve clarity.
- A table summarizing SPR kinetic parameters (ka, kd, KD) would be helpful for comparison across different antibodies.
Reviewer 2 Report
Comments and Suggestions for Authors
In this study, Wang et al. examine the binding of CA125-specific antibodies to MUC16 tandem repeats using mass spectrometry, ELISA, and SPR. Authors suggest that antibody M11 binds to all tested repeats from a newer model of CA125, and antibody OC125 fails to recognize five. They also report that discrepancies observed between ELISA and SPR appear to stem from contact time differences. While the study presents an interesting question, several methodological concerns limit its reliability and impact on the study’s conclusions.
Key concerns:
- It is unclear whether the repeats used in this study are accurately expressed. The authors attribute differences in band migration to proteoforms with identical amino acid sequences. However, this assumption does not necessarily confirm epitope similarity, even if amino acid sequences appear conserved. Additionally, since the proteins are expressed in a bacterial system, they are unlikely to undergo phosphorylation or other post-translational modifications. This raises the question of which proteoforms the authors speculate could contribute to the observed migration differences despite identical sequences. To address this, the authors should analyze the composition of their repeat peptides and verify the accuracy of plasmid sequences to rule out potential expression from mixed pools. These uncertainties impact the reliability of ELISA and SPR results and the overall re-validation of the new model, necessitating a more rigorous approach that involves identifying a common sequence among repeats and testing it in ELISA for epitope characterization.
- Coomassie blue and Western blot data indicate that most repeats exhibit distinct fragment migrations from purified repeat proteins, suggesting the presence of improperly folded or impure proteins. Given the sensitivity of ELISA and SPR, improperly folded or impure proteins may present different epitopes, affecting binding outcomes. For SPR, the observed non-binding trend corroborates with the repeats with the higher migrating band, which could be due to insufficient purification rather than intrinsic binding properties. The chromatographic purification method used (25 kDa size fractionation) appears insufficient, as shown by SDS-PAGE validations. The authors should further purify the peptides using HPLC or an equivalent method and subsequently verify sequence similarity and re-evaluate affinity data.
- The manuscript lacks clarity regarding the reproducibility of experiments. It is unclear whether the experiments were performed only once with three technical replicates or if they were conducted as independent biological replicates. If the data represent only technical replicates from a single purification batch, the authors should repeat the assays with at least three independent purifications and provide corresponding gel images to ensure data reliability.
- In Table 1, presenting sequence similarity within repeated regions and including a multiple sequence alignment of all repeats would be valuable. The authors should also compare protein sequences obtained via long-read sequencing and mass spectrometry-based sequencing. In ELISA experiments, bar plots with raw chemiluminescence values would allow better cross-antibody comparisons.
- In the discussion, the authors mention that "old R6" is similar to the new R5. If this is based on previous findings, a citation should be included, otherwise, the authors should provide supporting data. The claim that the study provides amino acid-level precision for antibody binding is premature, as definitive evidence identifying which amino acids are recognized by the four antibodies is not provided. This could be addressed by comparing the amino acid sequences of binding repeats and validating the findings experimentally.
Overall, while the study explores an important topic, the current manuscript lacks sufficient depth in analysis and methodological rigor. Additionally, it does not significantly advance the understanding of CA125 in ovarian cancer or contribute to improving current detection methods. Given these limitations, the manuscript, in its current form, does not appear to be a good fit for Cancer.
Reviewer 3 Report
Comments and Suggestions for Authors
Summary
This study investigated the binding patterns of CA125-specific antibodies (OC125, M11, and their clones, OC125-like and M11-like) to the tandem repeat region of MUC16. The study leveraged a revised molecular model of MUC16, enabled by long-read sequencing, to express and purify 16 of 19 MUC16 tandem repeats expressed in E. coli. The binding interactions of these proteins with four antibodies (OC125, M11, and their respective clones, OC125-like and M11-like) were characterized using ELISA and SPR. According to ELISA, the results reveal that M11 bound to all 16 repeats and M11-like bound all except R1, while OC125 bound to only 11 of the 16 repeats and OC125-like bound all except R11 and R17. SPR results reveal that M11 Bound 12 of 16 repeats (missing R4, R6, R9, R12) and OC125 bound 9 of 16 repeats. M11-like and OC125-like exhibited same pattern as ELISA. The study highlights discrepancies in binding patterns between ELISA and SPR, attributing them to differences in contact time and sensitivity between the two methods. The findings suggest that the current clinical CA125 test, which relies on both OC125 and M11 antibodies, may undercount MUC16 in patient serum due to the limited binding of OC125.
Major concerns:
- The study includes 16 out of the 19 tandem repeats in the MUC16 domain, with some repeats (R14, R18, R19) excluded due to challenges with protein solubility. While this is mentioned, it is recommended to describe any efforts made to address this issue or suggest alternative approaches that could be taken to resolve it.
- This study lacks exploration of biological consequences of differential binding (e.g., Are some epitopes more accessible in patient tumors? Are they glycosylated differently?)
No demonstration of how this data impacts patient samples, diagnostics, or treatment decisions. Please consider adding more details in the intro or discussion section.
- This study lacks of structural modeling, consideration of glycosylation, and in vivo or patient sample validation, which limits mechanistic or biological depth.
- While this manuscript is submitted for a special issue focusing on Ovarian Cancer Biomarkers, Diagnostics, and Therapeutic Technologies, it should elaborate further on the implications for the accuracy of CA125-based ovarian cancer diagnostics (e.g., Could patient undercounting occur? Whether the differential binding patterns could lead to false negatives in clinical testing?...etc.) in either discussion or intro section, especially since this study does not include in vivo or patient sample validation. The connection between differential binding patterns and their direct impact on clinical applications could be explored more deeply. Specifically, the manuscript could benefit from a clearer link between these findings and potential improvements in diagnostic sensitivity to enhance its significance.
Minor Concerns:
- Ref[8] and [19] are the same. Please either use [8] for both or revise [19] to a different citation.
Round 2
Reviewer 2 Report
Comments and Suggestions for Authors
Authors have addressed all my queries.